# Do Epilepsy Patients with Cognitive Impairment Have Alzheimer’s Disease-like Brain Metabolism?

**DOI:** 10.3390/biomedicines11041108

**Published:** 2023-04-06

**Authors:** Michael He, Tiffany A. Kolesar, Andrew L. Goertzen, Marcus C. Ng, Ji Hyun Ko

**Affiliations:** 1Undergraduate Medical Education, Rady Faculty of Health Sciences, University of Manitoba, Winnipeg, MB R3E 0T6, Canada; 2Department of Human Anatomy and Cell Science, Rady Faculty of Health Sciences, University of Manitoba, Winnipeg, MB R3E 0J9, Canada; 3Neuroscience Research Program, Kleysen Institute for Advanced Medicine, Health Sciences Centre, Winnipeg, MB R3E 3J7, Canada; 4Section of Nuclear Medicine, Department of Radiology, Rady Faculty of Health Sciences, University of Manitoba, Winnipeg, MB R3T 2N2, Canada; 5Graduate Program in Biomedical Engineering, Price Faculty of Engineering, University of Manitoba, Winnipeg, MB R3T 5V6, Canada; 6Section of Neurology, Department of Internal Medicine, Rady Faculty of Health Sciences, University of Manitoba, Winnipeg, MB R3E 0W2, Canada

**Keywords:** epilepsy, Alzheimer’s disease, FDG-PET, machine learning, support vector machine, metabolic classification, neurodegenerative disease, biomarker

## Abstract

Although not classically considered together, there is emerging evidence that Alzheimer’s disease (AD) and epilepsy share a number of features and that each disease predisposes patients to developing the other. Using machine learning, we have previously developed an automated fluorodeoxyglucose positron emission tomography (FDG-PET) reading program (i.e., MAD), and demonstrated good sensitivity (84%) and specificity (95%) for differentiating AD patients versus healthy controls. In this retrospective chart review study, we investigated if epilepsy patients with/without mild cognitive symptoms also show AD-like metabolic patterns determined by the MAD algorithm. Scans from a total of 20 patients with epilepsy were included in this study. Because AD diagnoses are made late in life, only patients aged ≥40 years were considered. For the cognitively impaired patients, four of six were identified as MAD+ (i.e., the FDG-PET image is classified as AD-like by the MAD algorithm), while none of the five cognitively normal patients was identified as MAD+ (χ^2^ = 8.148, *p* = 0.017). These results potentially suggest the usability of FDG-PET in prognosticating later dementia development in non-demented epilepsy patients, especially when combined with machine learning algorithms. A longitudinal follow-up study is warranted to assess the effectiveness of this approach.

## 1. Introduction

Alzheimer’s disease (AD) and epilepsy are two neurological disorders that have classically been considered separately from each other in clinical practice. AD is a chronic, progressive, neurodegenerative disease typically seen in the elderly and is the most prevalent cause of dementia [1]. Typical features of AD include neurocognitive impairment, memory loss, social withdrawal and changes in mood [2]. Epilepsy is a chronic neurological disease with a diverse set of etiologies comprised of recurring seizures [3], with clinical symptoms based on the brain regions and networks involved [4]. Postictal (i.e., post-seizure) symptoms can include confusion, agitation, and drowsiness, while long-term complications can include cognitive deficits [5,6] and psychiatric problems [7]. Temporal lobe epilepsy (TLE), the most common focal epilepsy [8,9], can lead to prominent long-term complications, including deficits in memory, language, and executive functioning [10].

The presence of either AD or epilepsy significantly increases the risk of the presence or development of the other, with AD patients having about a sixfold increased risk of developing epilepsy, and conversely, patients with epilepsy having at least double the risk of developing dementia [11,12,13,14]. This relationship likely works in a bidirectional way, with the disease process of one worsening the other and creating a negative feedback loop [13,15]. In a recent study by Lam et al. [16], intracranial foramen ovale electrodes were used to detect clinically silent seizures originating from the hippocampus in patients with AD for the first time. These seizures would normally be undetectable using conventional scalp electroencephalography. Furthermore, studies suggest that chronic hippocampal damage resulting from epileptic seizures can lead to cognitive impairment, memory loss, and an increased risk of dementia over time [17,18]. These studies provide further evidence that AD and epilepsy can occur concomitantly and that the neurophysiological signs, e.g., silent seizures in AD and AD-related pathological features in epilepsy that can be detected by positron emission tomography (PET) scans, may manifest subtly in the early stage. One important distinction between AD and epilepsy is that AD is typically diagnosed much later in life, with symptoms typically first appearing after the age of 65—although the underlying brain damage may begin decades earlier [19]—while epilepsy can be diagnosed as early as infancy and childhood [20]. Additionally, aging is the greatest risk factor for developing AD: two of the most characteristic pathological phenotypes in AD include neurofibrillary tangles and amyloid plaques, both of which can also be observed in greater quantity in normal aging [21].

Certain disease processes common to both AD and epilepsy are suggested to contribute to the underlying mechanism and may present similarly on functional brain imaging. In particular, chronic seizures in epilepsy may result in brain damage, due to neuronal degeneration and atrophy, represented by glucose hypometabolism on fluorodeoxyglucose (FDG) PET scans. This damage may ultimately lead to earlier or greater risk of dementia [22]. PET imaging can be used for the identification of epileptogenic zones [23], as well as possibly for the early diagnosis of dementia in people with mild cognitive impairment (MCI) [24]. FDG-PET often demonstrates lobar-specific patterns of hypometabolism that suggest particular underlying pathologies like AD [23]. AD is associated with areas of hypometabolism in the parietotemporal, posterior cingulate, hippocampal, and medial temporal regions, as well as the frontal lobe in later disease [25,26]. In epilepsy, there is high concordance between areas of aberrant FDG metabolism and seizures and epileptiform discharges [27,28]. In patients with epilepsy with long-term brain damage from chronic seizures, sclerotic changes can be found at the epileptogenic focus, e.g., temporal lobe sclerosis from temporal lobe epilepsy, leading to regional hypometabolism [29]. Work is ongoing to develop an objective biomarker using FDG-PET in AD and in epilepsy; however, there is currently no objective standard for its use as a quantitated marker in either disease.

In our previous study, we developed an automated FDG-PET reading algorithm using machine learning techniques that discriminated between patients with AD and age-matched healthy controls (machine learning-based AD designation; MAD) [30]. We tested five different algorithms and identified the support vector machine iterative single data algorithm (SVM-ISDA) as the strongest performing algorithm tested in differentiating those with AD from healthy controls, with a sensitivity of 0.84 and specificity of 0.95 in 10-fold cross-validation [30]. SVM is an effective algorithm [31,32] commonly used for neuroimaging [31,33], and uses a binary classifier that draws a boundary between different groups in higher-dimensional space. We have also demonstrated that MAD is also sensitive to other dementia types that are commonly seen in memory clinics, including dementia with Lewy bodies and primary progressive aphasia, and to a lesser degree, frontotemporal dementia [30,34]. Based on the congruence between AD and epilepsy, we applied the SVM-ISDA algorithm to assess the FDG-PET scans of patients with epilepsy and classify them as either MAD+ (AD-like) or MAD- (non-AD-like), particularly noting scans of patients with cognitive deficits as already determined by clinical neuropsychological assessments. Additionally, because of the importance of age in the pathology of AD, we only considered patients older than 40. Age 40 was decided on as a cutoff because although first symptoms often appear starting around age 65, neurological changes typically occur approximately 20 years earlier [35].

## 2. Materials and Methods

This study was a retrospective chart review. The patient sample consisted of 20 patients with epilepsy referred to the seizure clinic at the Health Sciences Centre in Winnipeg, Canada. Each patient had a PET scan between 2014 and 2019. Patient chart data were extracted from their electronic chart on Accuro electronic medical records software, including their age at date of scan, sex, diagnosis, medications, comorbidities, PET data, and neuropsychological testing results. Neuropsychological testing was conducted for 11 out of the 20 patients included in this study. Chart notes were not included as to why 9/20 patients were not referred to a neuropsychologist; however, it is possible that the treating clinician deemed this testing unnecessary due to no obvious signs of cognitive decline. Ethics approval for chart and image retrieval was obtained from the Biomedical Research Ethics Board at the University of Manitoba and Health Sciences Centre, in compliance with the Personal Health Information Act (PHIA).

Summaries of neuropsychological assessments were evaluated for evidence of cognitive deficits. An example of summaries suggestive of cognitive deficits includes “Behavioral evidence of executive functioning difficulties including perseverative tendencies, executive dyscontrol of memory, impulsivity, disinhibition and disorganization.” An example of summaries not suggestive of cognitive deficits includes “Mr. X’s profile was not suggestive of a general decline in cognitive functioning.” The assessments involved in this study were carried out and interpreted by the same neuropsychologist and consisted largely of the same battery of tests (i.e., Montreal Cognitive Assessment [MoCA] [36], Mini-Mental State Examination [MMSE] [37], Wechsler Abbreviated Scale of Intelligence [WASI-II], Wechsler Adult Intelligence Scale, fourth edition [WAIS-IV], Symbol Digit Modalities Test [SDMT], Brief Visuospatial Memory Test—Revised [BVMT-R], and digit span [DGS] tests).

Following a minimum 6-h fast, FDG-PET images were acquired using a Siemens Biograph 16 HiRez PET/computed tomography (CT) scanner (Siemens Medical Solutions, Knoxville, TN, USA) located in the John Buhler Research Centre at the University of Manitoba’s Bannatyne campus. Images were captured using standard T20s topogram settings. Patients were injected with 185 MBq of [^18^F]-FDG-PET, and a 15-min static image was acquired 40 min post-IV injection. A head CT scan was also acquired for attenuation correction.

PET data were preprocessed and analyzed in SPM12 software (version 7487, RRID:SCR_007037; Wellcome Trust Centre for Neuroimaging, London, UK, https://www.fil.ion.ucl.ac.uk/spm/(accessed on 19 February 2020)) using MATLAB (version r2018a, The MathWorks Inc., Natick, MA, USA), as described elsewhere [30,34]. Briefly, FDG-PET images were spatially normalized to standard MNI space for PET (using “Old Normalize” as this does not require T_1_-weighted images), then smoothened with an 8 mm full-width half-maximum Gaussian smoothing kernel. After preprocessing, the PET images were run through the SVM-ISDA machine learning algorithm (version 1, The Ko Lab, Winnipeg, MB, Canada, https://www.kolabneuro.com/software1/; accessed on 8 August 2020) in MATLAB to determine their individual MAD as either AD-like (MAD+) or not AD-like (MAD-). After the MAD analysis, an exploratory contrast (*p* < 0.01, uncorrected) was performed in SPM12 comparing the PET scans of cognitively normal epilepsy patients (+1; *n* = 5) to the PET scans of cognitively impaired epilepsy patients (−1; *n* = 6) to investigate metabolic differences between these patient groups. Finally, differences in the number of patients (MAD+ vs. MAD-) by cognitive status (cognitively normal, cognitive deficits, and unassessed) were examined using a chi-squared test. Statistical analysis was performed using IBM SPSS Statistics (version 25.0, IBM Corp., Armonk, NY, USA, released 2017) software.

## 3. Results

FDG-PET scans were included from a total of 20 patients with epilepsy, ranging from 40.9 to 69.9 years of age (see Table 1 for demographic data). Neuropsychological assessments were included, as available (see Table 1). All MAD designations were assigned as either MAD+ (AD-like) or MAD- (not AD-like; see Figure 1 for a visual representation of the “AD-like” metabolism represented by the reconstructed hyperplane that distinguished the two groups). Patients were on the following medications: lamotrigine (*n* = 13), levetiracetam (*n* = 12), brivaracetam (*n* = 1), divalproex (*n* = 3), valproic acid (*n* = 1), clonazepam (*n* = 2), lorazepam (*n* = 1, several times per month), clobazam (*n* = 4), lacosamide (*n* = 3), topiramate (*n* = 3), carbamazepine (*n* = 3), carbamazepine CR (*n* = 2), eslicarbazepine (*n* = 1), perampanel (*n* = 2), phenobarbital (*n* = 1), and phenytoin (*n* = 1). Although valproic acid, divalproex, and benzodiazepines have controversially been linked with an increased risk for developing AD, in the present study they do not appear to be linked with MAD designation [38,39,40,41].

Five patients were identified as MAD+, while the remaining 15 patients were identified as MAD-. A total of four out of six cognitively impaired patients were identified as MAD+, while all five cognitively normal patients were identified as MAD- (χ^2^ = 8.148, *p* = 0.017). For unassessed patients, one out of nine was identified as MAD+. The classification results are summarized in Table 2.

The exploratory contrast (*p* < 0.01, uncorrected) comparing the metabolism of cognitively normal epilepsy patients (+1, *n* = 5) to cognitively declined epilepsy patients (−1, *n* = 6) yielded no significant differences.

## 4. Discussion

The presence of MAD+ classifications in this study indicates that the disease process in epilepsy may result in brain metabolic patterns that resemble those found in AD. These metabolic patterns identified in the literature and our previous machine learning work [30] include hypometabolism in the posterior cingulate cortex/precuneus, parahippocampal gyrus, posterior parietal cortex, and bilateral and middle inferior temporal gyri, as well as hypermetabolism in the somatosensorimotor areas, basal ganglia, thalamus, and cerebellum [30,42,43,44,45]. In our current work, of the patients with epilepsy showing signs of cognitive decline, 67% (i.e., four out of six) were classified as MAD+. One of these patient’s clinical notes stated “suspected dementia due to AD or dementia with Lewy bodies,” although no formal diagnosis was made. While these results are interesting on their own, it is also important to note that none of the patients that had normal neuropsychological assessments were classified as MAD+. One patient out of nine from the “unassessed” group was designated as MAD+. Unfortunately, the clinical notes for this patient did not include rationale for why no neuropsychological assessment was performed, or whether this patient required a neuropsychological assessment and/or exhibited cognitive deficits at a later point in time. Although a larger sample is needed to confirm these findings, our recent work also showed low levels of MAD+ designations in healthy controls, those with stable MCI (not progressing to AD for >3 years) [30,34], and primary psychiatric disorders [34].

In any case, it is important to note that only a subset of patients with epilepsy will go on to develop AD; therefore, we do not expect all patients with epilepsy to develop AD. Importantly, we do not expect all patients exhibiting cognitive deficits to show AD-like hypometabolism either. However, this study is in line with previous reports that epilepsy patients are at higher risk of developing dementia [11,12,13,14,15] and that it may be detectable by FDG-PET scans using MAD or other automated algorithms. Although the current study highlights epilepsy and the potential for these patients to be at increased risk of developing AD, this work has possible implications for the clinical management for both diseases. For example, patients with AD who exhibit epileptiform activity often show cognitive decline years earlier than their nonepileptic AD counterparts [22].

The utility of this tool will likely be maximized in the context of the clinical setting where some cognitive deficits are present. Although a larger sample is needed to confirm these findings, it appears that only a subset of the patients with epilepsy experiencing cognitive decline exhibited AD-like brain metabolism. Future research is necessary to investigate whether patients who are identified as MAD+ continue on to develop AD and/or dementia, and if so, how long after a MAD+ identification occurs is a diagnosis typically made. Additionally, it will be important to follow up with the MAD- participants to assess whether or not they become MAD+ over time, whether or not they develop dementia, or if their cognitive impairment remains fairly stable. Ideally, this SVM-ISDA classifier will be applied to PET scans acquired during routine diagnostic testing in those experiencing cognitive decline to prognosticate the patient’s future outcomes. Although it appears that MAD can only identify AD-like brain metabolism in patients already experiencing some cognitive decline, this has real-world practicality. Brain PET scans are typically reserved for patients experiencing some sort of neurological symptoms, meaning that otherwise-healthy individuals need not acquire any unnecessary risks (e.g., radiation exposure, a false-positive MAD score), yet patients experiencing changes to cognition may still be identified early enough to make a difference in their disease course. Early diagnosis in this patient population is particularly important because people with both seizures and AD have a much worse disease course in terms of rate of cognitive decline, and earlier death [46]. Thus, identifying individuals with epilepsy at risk of developing AD early in their disease course can hopefully lead to disease-modifying treatment to improve prognosis.

Interestingly, the contrast directly comparing brain metabolism between cognitively normal and cognitively impaired patients with epilepsy yielded no significant results. Although the sample was small (*n* = 11), these data are important in the larger context of this work: the MAD algorithm was able to identify subtle differences in brain metabolism among these patients, while a grouped comparison was not. These nonsignificant contrast results highlight the possible utility and power of the MAD designation in the clinical setting. Individualized medicine becomes a more realistic target as tools such as the MAD algorithm improve to the point of being able to identify aberrant metabolism at the individual level, especially when these differences cannot be identified in typical group-level contrasts.

Although recent machine learning work using magnetic resonance imaging to distinguish neural structural differences between epilepsy and AD has begun [47], to our knowledge, this is the first study to use machine learning to investigate how AD-like brain metabolism can be observed in epilepsy. While there are, of course, many differences between these diseases, there are also some important similarities. Although the pathophysiology of both AD and epilepsy are debated, there is considerable overlap in the hypotheses for the brain damage that results from each of these diseases. For example, a variety of hypotheses for the degeneration observed in AD include amyloid/tau accumulation [48,49], cholinergic/oxidative stress, glucose hypometabolism [50], and glutamate excitotoxicity [51]. For epilepsy, hypotheses include neuronal damage and subsequent neurodegeneration caused by either the underlying processes causing seizures [52] or the seizures themselves, which may cause oxidative stress [53], as well as underlying genetic etiology such as tuberous sclerosis [54] or Rett syndrome [55]. Recent evidence also points to network dysfunction and excitotoxicity in epilepsy [56,57] and establishes links to β-amyloid and hyperphosphorylated tau accumulation [46,58,59], such as that seen in AD. Other contributors to both disease processes may include common vascular comorbidities [15], and astrocytic glutamate homeostasis dysregulation [60,61]. Importantly, the cognitive deficits commonly observed in epilepsy are also similar to those observed in the prodromal stage of AD (i.e., MCI), such as executive dysfunction and language deficits, and particularly memory loss in TLE [6,10]. Furthermore, recent evidence suggests a common genetic and molecular connection between epilepsy and AD, relating to calcium signaling and binding [62]. Although more research is needed to determine which patients with epilepsy will and which will not go on to develop dementia or AD, the present work provides a possible means for distinguishing between these patients.

While the MAD algorithm shows much promise for early diagnosis of AD, this study does have some drawbacks. As mentioned, half of the patients with epilepsy were not formally assessed by a neuropsychologist. It is possible that these patients were not referred to see a neuropsychologist because the treating clinician did not notice any signs of cognitive decline, e.g., only one of nine unassessed patients was identified as MAD+. Additionally, this study is limited in terms of sample size and the lack of a long-term follow-up diagnosis of dementia. Important future research stemming from this work should include a longitudinal study to assess the accuracy of these MAD+ designations for predicting future AD development in patients with epilepsy. Use of machine learning in this context may provide care providers and patients with a prognostic biomarker for developing AD in the future and may allow for the use of early interventions to slow disease progression.

The current data add to the mounting evidence for AD-like neuropathology in epilepsy, which has potential implications for how each disease should be considered in a clinical setting. Early and effective treatment for each disease may help slow the added deleterious effects that the combination of diseases causes when both are present. While patients with AD who develop epilepsy tend to respond better to lamotrigine and levetiracetam than phenytoin for preventing seizures [22], earlier identification of AD in patients with epilepsy may also provide the opportunity for treatment with anti-AD medications such as aducanumab [63].

## 5. Conclusions

In this study, we applied an SVM-ISDA machine learning algorithm trained on the FDG-PET scans of AD patients and healthy controls to those of patients with epilepsy in the context of pathophysiological and clinical overlap between the two diseases. To our knowledge, this is the first machine learning study to include functional imaging using FDG-PET to investigate the brain metabolic pattern similarities of AD and epilepsy. Cognitively normal patients were all designated as MAD-. These results suggest that cognitively normal patients with epilepsy do not exhibit AD-like hypometabolism, confirming the high specificity of the SVM-ISDA algorithm. Additionally, two-thirds of the cognitively impaired patients with epilepsy were identified as MAD+. These results potentially suggest a usefulness of FDG-PET and MAD in identifying dementia risk in epilepsy at the individual level, while a larger-scale study with longitudinal follow-up is warranted.

## Figures and Tables

**Figure 1 biomedicines-11-01108-f001:**
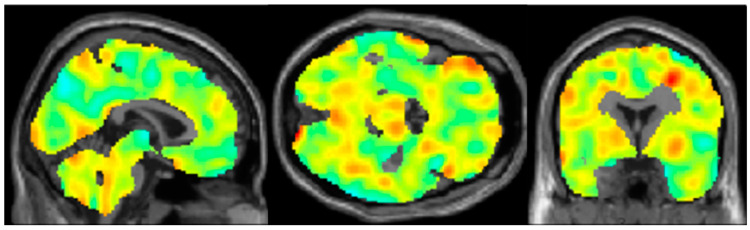
Visualization of the reconstructed hyperplane maximally separating AD patients from healthy controls using the SVM-ISDA algorithm, developed from our previous work [30]. Warmer colors (red, orange) indicate regions of hypermetabolism, while cooler colors (blue, green) indicate regions of hypometabolism. PET scans with a high degree of concordance in metabolism with this hyperplane are designated as MAD+ and highly discordant scans are designated as MAD-.

**Table 1 biomedicines-11-01108-t001:** Demographic data, including mean age in years (±SD), sex, epilepsy localization, and cognitive status, as available.

Demographic		
*N*		20
Mean age (±SD)		50.9 (±9.4)
Sex (female:male)		10:10
Epilepsy localization (*n*)		
	Temporal lobe	6
	Frontal lobe	2
	Frontotemporal lobe	2
	Parietal/occipital lobe	1
	Other ^1^	9
Cognition (*n*)		
	Not assessed	9
	Cognitively impaired	6
	Cognitively normal	5

^1^ Examples of “other” epilepsy localization largely include epilepsy of undetermined origin or localization occurring across several regions.

**Table 2 biomedicines-11-01108-t002:** Positive machine learning-based Alzheimer’s disease designation (MAD+, i.e., Alzheimer’s disease-like) designations. Results are broken down by neuropsychological assessment, including patients who did not receive an assessment, patients who were assessed to have cognitive deficits, and patients who were assessed as cognitively normal.

	MAD+ (*n*)
All participants	5/20
Not assessed	1/9
Cognitively impaired	4/6
Cognitively normal	0/5

## Data Availability

The MAD algorithm is available via www.kolabneuro.com (accessed on 8 August 2020). Patient files are not available as per the Patient Health Information Act.

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
