# Peer review of "Do Epilepsy Patients with Cognitive Impairment Have Alzheimer’s Disease-like Brain Metabolism?"

_biomedicines, 2023, doi:10.3390/biomedicines11041108_

Round 1

Reviewer 1 Report

In this manuscript, authors presented an interesting topic. They investigated if patients with epilepsy who show mild cognitive symptoms also show AD-like metabolic patterns using their previously developed automated fluorodeoxyglucose positron emission tomography (FDG-PET) machine learning-based AD designation (MAD) reading program. Their results suggested a usability of FDG-PET in prognosticating later dementia development in non-demented epilepsy patients, especially when combined with machine learning algorithms.

Major point:

Nine out of 20 patients were not assessed for the cognitive deficits. If the aim of study is to investigate whether patients with epilepsy who show mild cognitive symptoms show AD-like metabolic patterns, what is the reason for not assessing the cognitive functions in these patients? If 45% of study subjects were not assessed for the cognitive impairment, this affects the objective of study and interpretation of data. Proper study design should have included a selection criterium of assessment of cognitive functions in all the participants.

I see that the authors interpreted their findings in the discussion based mainly on the data in the 6 patients having epilepsy with cognitive decline and the data from the 11 patients (comparing brain metabolism between cognitively normal and cognitively impaired patients with epilepsy). This prevents the inaccurate interpretation of data, I agree with that.

The authors mentioned a possible reason for not assessing the 9 patients in the drawbacks in the discussion section, I suggest moving this reason to the materials and methods section. I also suggest indicating that in the abstract as following; line 24: “In this study, we investigate if epilepsy patients with/without mild cognitive symptoms also show AD-like metabolic patterns determined by the MAD algorithm”.

Minor point:

Figure 1: doesn’t explain the images provided in the figure, please add brief explanation of the figure.

Reviewer 2 Report

Dear Authors,

I have read the manuscript entitled "Do Epilepsy Patients with Cognitive Impairment have Alzheimer's Disease-like Metabolism?".

The original work, based on a clinical trial, focuses on studying the identification of similarities in brain metabolism in patients with epilepsy and those with Alzheimer's dementia.

The manuscript complies with the requirements of the journal, the bibliographic references are appropriate for the presented topic. I appreciate that you also gave us the Conclusions section. The table and figure inserted in this material improve the reading and make it easier to understand some aspects.

However, I have a number of questions and suggestions:

1. Could the authors improve the title? Since in the manuscript you referred to brain metabolism, as expected in these pathologies, it seems more appropriate to state the type of metabolism in the title.

2. In the Material and Methods chapter, you indicated that the study took place between 2014-2019. Over the 5 years, how many scans were performed on each patient?

3. Since we are talking about epileptic patients, what therapy did these patients follow?

4. It is well known that some drug classes used as anticonvulsant therapy (benzodiazepines, valproic acid) can increase the risk of Alzheimer's Dementia. Can the authors argue that in the case of the 20 patients, the therapy may have caused neurodegenerative damage?

5. In the Discussions chapter, the first sentence “The presence of MAD+ classifications in this study indicates that the disease process in epilepsy may result in brain metabolic patterns that resemble those found in AD”, could it be detailed? Describe what hypometabolism refers to.

6. The number of patients enrolled in the study is a bit small. Are the statistical results convincing enough?

7. The English language could be improved.
